# Aluminum Oxide Ceramic Coatings on 316l Austenitic Steel Obtained by Plasma Electrolysis Oxidation Using a Pulsed Unipolar Power Supply

**Victor Aurel Andrei** [1,2]**, Cristiana Radulescu** [2,3,*]**, Viorel Malinovschi** [4]**, Alexandru Marin** [5]**,
Elisabeta Coaca** [5]**, Maria Mihalache** [5]**, Cristian Nicolae Mihailescu** [6]**, Ioana Daniela Dulama** [2]**,
Sofia Teodorescu** [2] **and Ioan Alin Bucurica** [2]

[1]    ELSSA Laboratory SRL, 110109 Pitesti, Romania; andvic12@yahoo.com
[2]    Valahia University of Targoviste, Institute of Multidisciplinary Research for Science and Technology,
       130004 Targoviste, Romania; dulama_id@yahoo.com (I.D.D.); sofiateodorescu@yahoo.com (S.T.);
       bucurica_alin@yahoo.com (I.A.B.)
[3]    Faculty of Sciences and Arts, Valahia University of Targoviste, 130004 Targoviste, Romania
[4]    Department of Environmental Engineering and Applied Engineering Sciences, University of Pitesti,
       110040 Pitesti, Romania; viorel.malinovschi@gmail.com
[5]    Institute for Nuclear Research Pitesti, POB 78, Mioveni, Romania; marin.alexandru.horia@gmail.com (A.M.);
       elisabeta.coaca@nuclear.ro (E.C.); maria.mihalache@nuclear.ro (M.M.)
[6]    National Institute for Lasers, Plasma and Radiation Physics, 077125 Magurele, Romania;
       cristi.mihailescu@inflpr.ro
[*]    Correspondence: radulescucristiana@yahoo.com; Tel.: +40-245-206-109

**Abstract:** AISI 316 steel has good corrosion behavior and high-temperature stability, but often prolonged exposure to temperatures close to 700 °C in aggressive environments (e.g., in boilers and furnaces, in nuclear installations) can cause problems that lead to accelerated corrosion degradation of steel components. A known solution is to prepare alumina ceramic coatings on the surface of stainless steel. The aim of this study is to obtain aluminum oxide ceramic coatings on 316L austenitic steel, by Plasma Electrolysis Oxidation (PEO), using a pulsed unipolar power supply. The structures obtained by PEO under various experimental conditions were characterized by XPS, SEM, XRD, and EDS analyses. The feasibility was proved of employing PEO in $NaAlO_2$ aqueous solution using a pulsed unipolar power supply for ceramic–like aluminum oxide films preparation, with thicknesses in the range of 20–50 μm, and a high content of $Al_2O_3$ on the surface of austenitic stainless steels.

**Keywords:** AISI 316 steel; $Al_2O_3$; plasma electrolysis oxidation; X-ray photoelectron spectroscopy; scanning electron microscopy; X-ray diffraction; energy dispersive x-ray spectrometry

## 1. Introduction

Chromium-nickel stainless steels are designed primarily for application in strong hot sulphuric acid solutions. The required corrosion resistance is achieved primarily through the use of high nickel content.

Austenitic stainless steel AISI 316L has a relatively low cost and high corrosion resistance, an excellent processability, and high temperature stability but has a low surface hardness and low wear resistance [1,2]. Its primary alloying constituents after iron, are chromium (between 16%–18%), nickel (10%–12%), and molybdenum (2%–3%). The addition of molybdenum provides greater corrosion resistance than AISI 304, with respect to localized corrosive attack by chlorides and to general corrosion

by reducing acids, such as sulfuric acid [3,4]. Stainless steel 316L grade is the low carbon version of stainless steel 316.

Due to its properties, AISI 316 steel is used in many areas, such as construction of exhaust manifolds, furnace parts, heat exchangers, jet engine parts, evaporators, chemical processing equipment, nuclear equipment, valve and pump parts, and parts that are exposed to marine environments [5–8]. The high temperature corrosion commonly occurs in boilers and furnaces, in most cases by the combustion of coal causing a chemical process, where most steels have their metal surface in combustion ashes whose composition consists of mixtures of sulphates and chlorides, generating problems such as decrease in heat transfer, which leads to corrosion in an accelerated manner [9,10]. Chromium-nickel steels ensure good performances above 1200 °C, due to their austenitic type structure; however, after being subjected to temperatures close to 700 °C in aggressive environments for prolonged periods, the oxidative attack that causes corrosion occurs. A known solution is to prepare alumina ceramic coatings on the surface of stainless steel [11–13].

Alumina is one of the important technical ceramics due to its useful properties (i.e., hardness, thermal, chemical, and dielectric) [14–16]. Alumina films have received considerable attention as high temperature engineering material [17]. There is an increasing technological need to protect metals in aggressive environments such as acidic or oxidizing environments [18]. In order to improve the corrosion behavior of AISI 316 steel at high temperatures, in aggressive environments, $Al_2O_3$ coatings were obtained by thermal spray (a technique of projecting droplets of molten material for coating surfaces of parts) [11,19]. Droplets of molten material are accelerated in a gas jet projecting against the surface to be coated. Thermal spraying means to cause a molten material to collide with and accumulate on the surface of a substrate [20,21]. Accordingly, in order to obtain a good thermal spray coating, a high-temperature is required to create an adequate molten phase as well as a high-speed for spraying particles. In the film-creation process by thermal spraying, molten particles collide with the substrate and simultaneously become flattened. Particles of the raw material that have melted and become a liquid build up rapidly to form a film.

The thermal spraying technique has some limitations related to the geometry of the substrate (low degree of adhesion on small substrates and substrates with small curvature) [19,21]. AISI 316L steel becomes a good choice of structural material for the fusion reactor in the future, owing to its excellent processability and high temperature stability, but the structural pipe fittings achieved by 316L stainless steel are easily corroded by a large amount of highly permeable and corrosive tritium in the operating environment of fusion reactor [22]. The world-wide acknowledged solution is to prepare alumina ceramic coatings on the inner surface of stainless-steel pipes, which cannot only guarantee the structure property of pipe systems, but also protect the pipes from the permeation and corrosion of tritium [12].

Austenitic 316L steel has already been extensively used as a nuclear structural material and is among the materials selected for nuclear systems with Heavy Liquid Metals (HLM) [5,23] as lead or Lead Bismuth Eutectic (LBE) [24]. The use of HLM raises problems with the compatibility of materials in terms of corrosion and mechanical strength [25,26]. Austenitic steels suffer from severe corrosion attack in lead or LBE melt at temperatures above 500 °C [24]. To improve the corrosion resistance of stainless steels exposed to HLM containing oxygen at temperatures above 500 °C, it is of interest to develop surface engineering techniques in order to alloy the steel surface with Al [25]. In order to improve the corrosion resistance of stainless steels exposed to oxygen-containing HLM at temperatures above 500 °C, a procedure MIEB-Al (Microsecond-pulsed Intense Electron Beams) was applied in order to alloy the steel surface with Al [4]. The procedure itself consists in two stages: (*i*) Al or Al alloy deposition and (*ii*) melting of deposited layer on the steel surface using intense pulsed electron beam. The thickness of the aluminum containing layer (10–30 μm) is around the penetration depth of electrons into the steel. Therefore, by applying the MIEB-Al procedure, the microstructural properties of the substrate materials (excepting the superficial layer) do not change and it is possible to obtain a surface layer with a uniform distribution of Al, controlled thickness, crack-free, and adherent to the substrate. There is interest in developing surface engineering techniques, less expensive and more

permissive with the geometry of treated components, better than the thermal spraying technique and the MIEB-Al procedure [4].

Plasma Electrolytic Oxidation (PEO) is a cost effective and environmentally-friendly surface engineering technique which is becoming more widely used to improve the surface properties of materials [27]. Compared with other techniques that can be used to make protective ceramic coatings on austenitic steels (the thermal spraying technique and the MIEB-Al procedure), PEO is cheaper and more suitable for treating complex geometry samples. The PEO is an advanced form of anodic oxidation and primarily applied to valve metals and their alloys, characterized by spontaneous formation of a barrier layer in contact with the electrolyte; the barrier layer is required for initiation of the micro-discharge oxidation processes [28,29].

In the case of non-valve metals (base metals) as steels, it can be used as two methods to develop a porous oxide barrier layer required for initiation of the micro-discharge oxidation processes [30–32]:

- applying a preliminary treatment: the deposition on the steel surface of Al or other valve metal as Ti or Zr layer by a certain method (e.g., a porous oxidation barrier layer necessary to initiate the PEO process can be developed on carbon steel by spraying with an aluminum [28], or by aluminizing using dipping technique [29], and in the case of stainless steel, the specimens can be coated with a Ti film by magnetron sputtering [33], or aluminum/bimetallic stainless steel may be used [34], then it can be obtained the porous oxide film in the anodic oxidation stage of PEO process); the oxidation in autoclave of the stainless steel, which produces a thick layer of magnetite over a substrate [30];
- deposition of the porous oxide ($Al_2O_3$, $SiO_2$) resulted from the micro-arc decomposition of aluminate or silicate electrolyte [35], in the case of carbon steel.

For stainless steels (particularly 316L steel) no data were reported about deposition of the porous oxide resulted from the micro-arc decomposition of aluminate or silicate electrolyte and our attempts were not successful.

Recently, the feasibility of using PEO in aqueous $NaAlO_2$ for preparation of ceramic–like aluminum oxide films on the surface of austenitic stainless steels was proved [32]. After preliminary treatment by the autoclave oxidation, which produces a thick layer of magnetite suitable for the formation of the barrier layer over the 316L steel substrate, ceramic-like layers with a thickness of ≈ 30 μm, made of aluminum oxide were obtained by micro-arc oxidation in 0.1 M $NaAlO_2$ and 0.05 M NaOH aqueous solution, using DC, constant voltage conditions, 320 V. Ceramic-like aluminum oxide coatings were non-uniform, porous, and discontinuous.

Improving the coatings quality and gaining its new physical and chemical properties could be carried out in two ways: by developing and use of new electrolyte compositions or applying special electric modes of the coating formation. Currently, in the field of the microarc oxidation technology there is a trend in using the pulse mode when processing. The aim of this study is to obtain aluminum oxide ceramic coatings on 316L austenitic steel, by Plasma Electrolysis Oxidation using a pulsed unipolar power supply.

## 2. Materials and Methods

### 2.1. Preparation of PEO Coatings

AISI 316L stainless steel samples ($20 \times 15 \times 1$) mm$^3$, were manually grounded up to 1200 mesh SiC paper to achieve a fine finish with an average surface roughness of 0.35 mm, which minimizes the mechanical surface damage and allows a good adherence of a coated layer [36]. Then the samples were cleaned with distilled water and ethanol before treatment. The test samples were cut from the sheet so they have a rod in extension to ensure samples fastening and the electrical contact. The rod was insulated in order to avoid its oxidation.

An advantage of the PEO technique is that it allows the treatment of samples with different shapes and sizes; the important parameter is the current density, the treatment of large samples requiring the use of high-power sources. The dimensions of the samples used in this study are suitable for the development of the treatment method, being suitable for performing different analyses to characterize the coatings made.

Samples were autoclaved in deionized water at 350 °C and 160 atm for 50 days. Plasma electrolytic oxidation of 316L samples was performed using a unipolar pulsed DC power source with 150 Hz frequency [37].

Electrolyte solutions containing sodium aluminate at concentrations 25 g/L were prepared using pure reagents and distilled water. The electrolyte has a pH = 13 and electrical conductivity 27.3 mS/cm. For the micro-arc oxidation process of 316L the potentiostatic regime has been used applying an effective voltage value of 260 V (S1) and 220 V (S2), respectively. The voltage actual value was slowly increased until the electric spark discharge ($U$sd) occurred, after which being rapidly increased to 260 V in the case of S1 sample and to 220 V in the case of S2 sample. The micro-arc treatment time ($\tau$) was set at 5 min and the electrolyte temperature was kept below 20 °C. Sample codes, voltage actual values spark discharge ($U$sd), applied effective voltages ($U$), impulse amplitudes ($U$ amp), duty cycles ($\eta$), and treatment time ($\tau$) can be found in Table 1.

**Table 1.** Sample codes and oxidation conditions during plasma electrolytic oxidation of 316L samples in aluminate electrolyte.

| Sample Code | $U$sd [1] [V] | $U$ [2] [V] | $U$amp [3] [V] | $\eta$ [4] [%] | $\tau$ [5] [min] |
|---|---|---|---|---|---|
| S1 | 120 | 260 | 580 | 40 | 5 |
| S2 | 120 | 220 | 540 | 37 | 5 |

[1] $U$sd—voltage actual values spark discharge, [2] $U$—applied effective voltages, [3] $U$amp—impulse amplitudes, [4] $\eta$—duty cycles, [5] $\tau$—treatment time.

## 2.2. Characterization of PEO Coatings

In order to understand the processes of electrolytic oxidation and to develop the experimental method for making ceramic coatings based on aluminum oxide on a stainless-steel substrate, it is necessary to characterize the deposits in terms of morphology, structure, and composition both in the surface area and in the bulk deposition; the properties of the surface area determine the corrosion behavior of the material.

X-ray Photoelectron Spectroscopy (XPS) analysis was carried out in order to determine the surface composition. The electron spectrometer (ESCALAB 250, Thermo Fisher Scientific GmbH, Dreieich, Germany) was used, which ensures an energetic resolution smaller the 0.45 eV for XPS technique and facilitates localized spectroscopy (spot less than 120 μm). XPS spectra were recorded using Al Kα monochromatized source (hv = 1486.6 eV) in a vacuum of $10^{-8}$ Pa. The acquired spectra were calibrated with respect of the C1s line of surface adventitious carbon at $E_B$ = 284.8 eV (where $E_B$ represents the binding energy of the electron with respect to the vacuum level). An electron flood gun was used to compensate the charging effect in insulating samples. The analyzed areas were cleaned by Ar ion beam etching. A delocalized $Ar^+$ ion beam accelerated under 2 keV was used to remove absorbed contaminants on the surfaces. Survey spectra were acquired in the following conditions: X-ray spot size = 500 μm, Number of Scans = 1, Pass Energy = 100 eV, Energy Step Size = 1 eV. High resolution spectra were acquired on energy regions for Al2p, O1s, Fe2p, Cr2p, Ni2p peaks under the following conditions: X-ray spot size = 500 μm, Number of Scans = 50, Pass Energy = 10 eV, Energy Step Size = 0.1 eV.

X-ray Diffraction (XRD) analysis was used to examine the crystal composition of the oxide layers employing a Rigaku diffractometer (Ultima IV, Rigaku, Woodlands, TX, USA) with Cu Kα radiation. For the qualitative phase analysis, the X-ray diffractograms were obtained by scanning the ($2\theta$) = 20° −

80° range with 0.05° step and 8 s step-time using Bragg-Brentano (θ – θ) focusing scheme and graphite monochromator in diffracted beam.

Scanning Electron Microscopy (SEM) and elements microanalysis (Cross-sectional SEM images and EDS analysis at selected points), using the electrons probe, was performed using a TESCAN Electron Microscope (VEGA II LMU, TESCAN, Brno, Czech Republic). SEM device was equipped with 3 detectors: a secondary electron (SE) detector, a back-scattered electron (BSE) detector, and energy dispersive x-ray spectrometer (EDS).

Analysis of surface morphology of PEO coatings and elemental analysis was performed using a scanning electronic microscope (SU-70, Hitachi, Tokyo, Japan) coupled with energy dispersive X-ray spectrometer (UltraDry, Thermo Fisher Scientific GmbH, Dreieich, Germany). The Thermo Scientific™ Pathfinder™ X-ray Microanalysis Software allows achieving the element distribution maps.

Potentiodynamic polarization measurements were performed on a potentiostat/galvanostat (PARSTAT-2273, Princeton Applied Research, Oak Ridge, TN, USA) in a three-electrode set, including the working electrode (test sample), a platinum electrode and a saturated calomel $Hg_2Cl_2$) (0.244 V vs. SHE at 25 °C). Polarization curves were obtained in 0.5 M NaCl aqueous electrolyte, at scan rate of 0.5 mV/s and in the range of −0.25 V ÷ +0.25 V versus open circuit potential, step height 1 mV. The corrosion behavior was investigated using Tafel slope method.

## 3. Results

### 3.1. X-ray Photoelectron Spectroscopy

Figure 1 shows the XPS survey spectra corresponding to sample autoclaved before PEO (a), as well as, overlapped spectra corresponding to samples S1, S2 (b). Elements highlighted in addition to common contaminants (C, Ca, Na, Cl, Ar due to Ar ion bombardment) are Al, O on sample S1 and Al, O, Fe on sample S2. Note the absence of Ni and Cr.

Figure 2 shows high resolution spectra for Al2p, O1s, and Fe2p peaks for S1 and S2 samples, respectively.

Fe present on surface S2 is valence 2 (FeO), as shown by the presence of the shake-up satellite in the valley between 2p3/2 and 2p1/2 component [37,38]. The results obtained show that the sample surface S1 is coated with $Al_2O_3$, and the surface of sample S2 is mostly coated with $Al_2O_3$, with FeO being present in a small amount. Elements highlighted in addition to common contaminants (C, Ca, Na, Cl, Ar due to Ar ion bombardment) are Al, O on sample S1, and Al, O, Fe on sample S2. Note the absence of Ni and Cr. The appearance of aluminum oxide (Al2s, Al2p and O2s) is highlighted on the surface of the treated sample (Table 2). Al2p peak has closely spaced spin-orbit components; splitting may be ignored for Al2p peaks from Al oxide. Aluminum oxide is an insulating material and oxide peak position may vary with the thickness of the film. The difference between the peak shape Al2p for sample S1 and for sample S2 highlighted in Figure 2a is explained by the difference in thickness of the oxide film in the case of the two samples. XPS measurements show clearly that on the surface of the sample S1 are not present the elements Fe, Cr, Ni but only aluminum oxide, and that on the surface of the sample S2 are not present the elements Cr, Ni but but aluminum oxide and iron oxide (lower quantity).

XPS analysis provide information about surface (50 Å depths): the top layer of the coating contains aluminum oxide and, in the case of sample S2, also contains a trace of FeO.

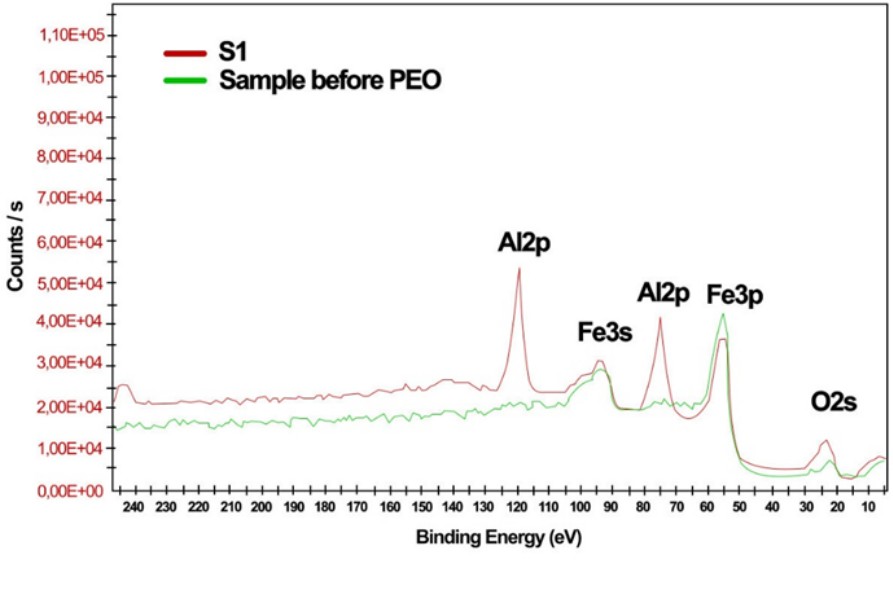

(**a**)

(**b**)

**Figure 1.** XPS survey spectra of: (**a**) XPS spectra for S1 sample compared with 316L sample autoclaved before PEO; (**b**) XPS survey spectra of 316 L samples after PEO treatment: S1 (red line) and S2 (blue line).

**Table 2.** XPS elemental composition of samples (S1 and S2).

| Sample Code | Name | Peak BE (Binding Energy) | Atomic [%] |
|---|---|---|---|
| S1 | Al2p | 74.6 | 35.8 |
| | O1s | 531.1 | 64.2 |
| | Al2p | 74.6 | 35.7 |
| S2 | Fe2p | 709.3 | 3.0 |
| | O1s | 531.1 | 61.3 |

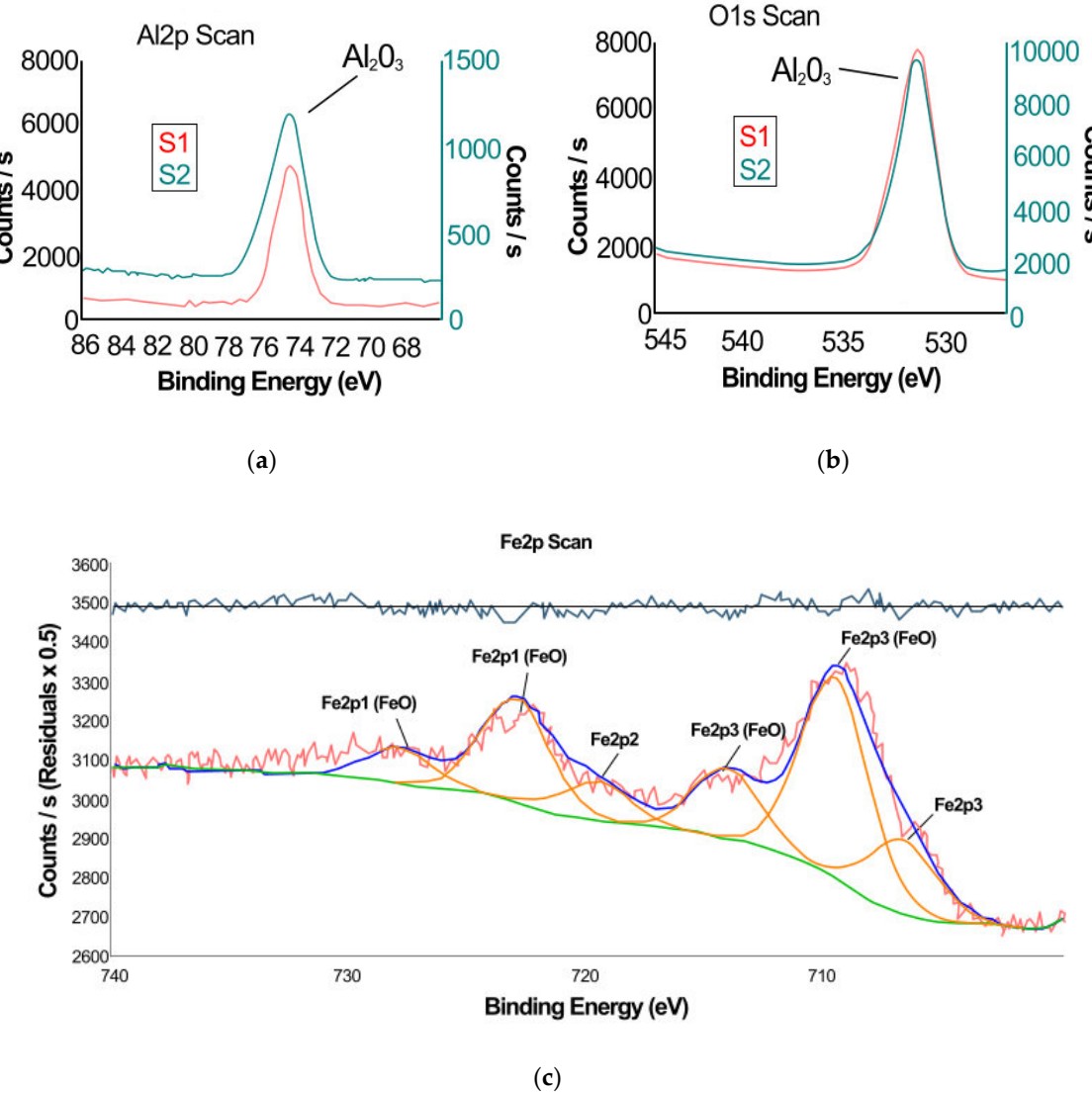

**Figure 2.** Overlapped XPS spectra of S1 and S2 sample: (**a**) Al2p; (**b**) O1s; (**c**) Fe2p.

### 3.2. X-ray Diffraction

Qualitative phase analysis results of the recorded XRD patterns of the samples in this study evidence the presence of polycrystalline phases, which were further distinguished as $Al_2O_3$ (JCPDS 89-7715), $FeAl_2O_4$ (JCPDS 86-2320), $Fe_2O_3$ - alpha (JCPDS 89-597), Fe (JCPDS 88-2324). Figure 3 shows the result of the qualitative phase analysis for the diffraction spectra acquired for both samples (S1 and S2).

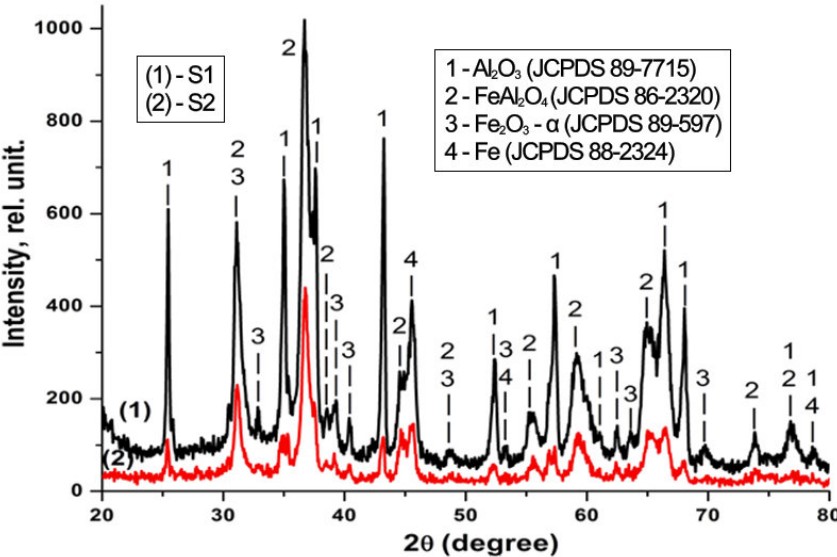

**Figure 3.** XRD patterns of PEO coatings formed in 25g/l NaAlO$_2$ electrolyte.

### 3.3. Morphology and Microstructure of PEO Coatings

SEM images of the surfaces of 316L stainless steel samples treated by PEO, at different magnifications, are shown in Figure 4. Figure 4a shows the magnetite crystallites formed by autoclaving.

The applied treatments lead to the formation of adherent structures. According to Figure 4, PEO treatment produces a porous top layer; SEM micrographs at low magnification show the presence of relatively large outside pore and open to the surface. As high magnification SEM micrographs show, PEO treatment leads to the formation of a nanostructured superficial layer.

Figure 5 exhibits the distribution of the elements on the surface of 316L stainless steel samples treated by PEO. Figure 5 shows the uniform distribution of Al, O, Fe elements on the surface of 316L stainless steel samples treated with PEO; distribution O follows the Al distribution, highlighting the formation of Al oxide, for both samples. The data corresponding to X-ray maps (Figure 5) are mentioned in Table 3, as average values of atom.% ± S.D.%.

Cross-sectional SEM images for S1 and S2 samples along with observed oxide layer thickness are shown in Figure 6. Figure 7 shows the results of the analysis at selected points in the section of the samples S1 and S2.

**Table 3.** Elemental content (EDS analysis) of samples.

| Element | S1 | S2 |
|---|---|---|
| | Content [atom.%±S.D.%] | |
| Al | 25.0 ± 0.09 | 30.0 ± 0.11 |
| O | 49.0 ± 1.42 | 57.0 ± 0.27 |
| Fe | 1.4 ± 0.07 | 0.7 ± 0.05 |
| Na | nd | 2.3 ± 0.02 |
| C | 24.6 ± 0.12 | 10.0 ± 0.15 |

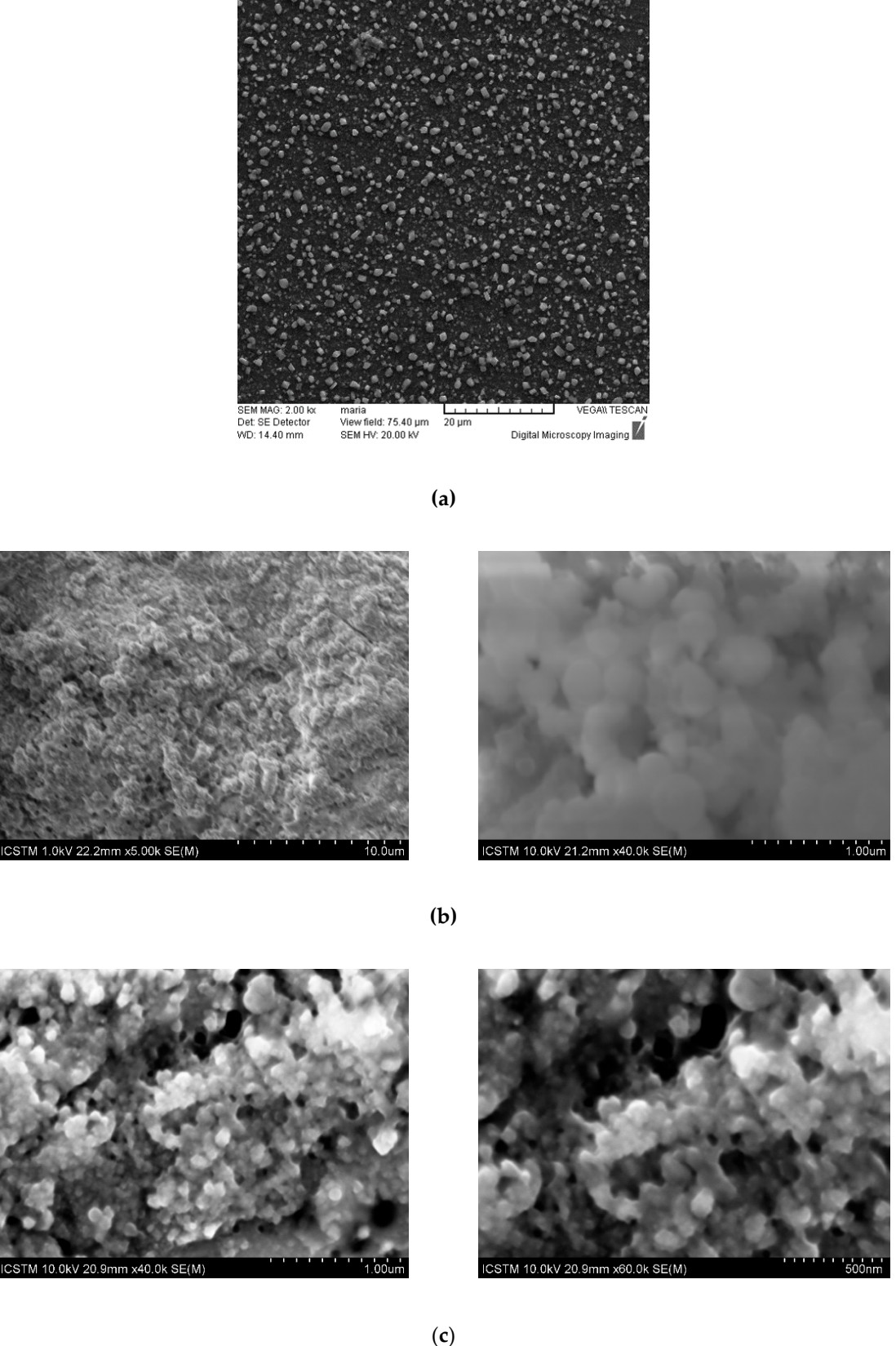

**Figure 4.** SEM images of the surfaces of 316 L stainless steel samples treated by PEO: (**a**) 316L autoclave sample, before PEO treatment; (**b**) S1 sample; (**c**) S2 sample.

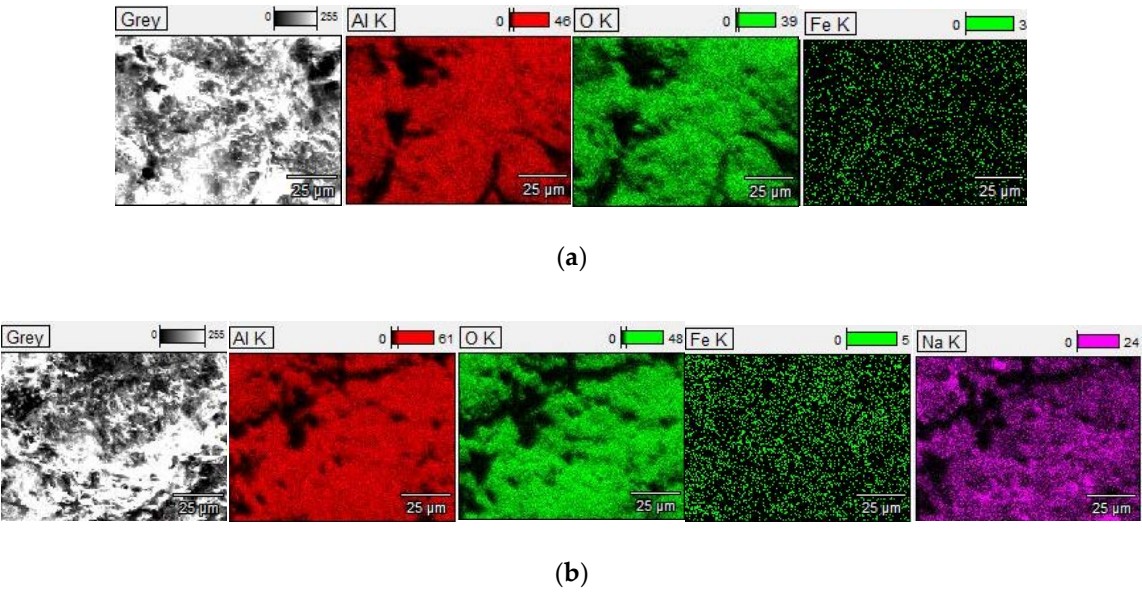

(**a**)

(**b**)

**Figure 5.** The distribution of the elements on the surface of 316L stainless steel samples treated by PEO:
(**a**) S1 sample; (**b**) S2 sample.

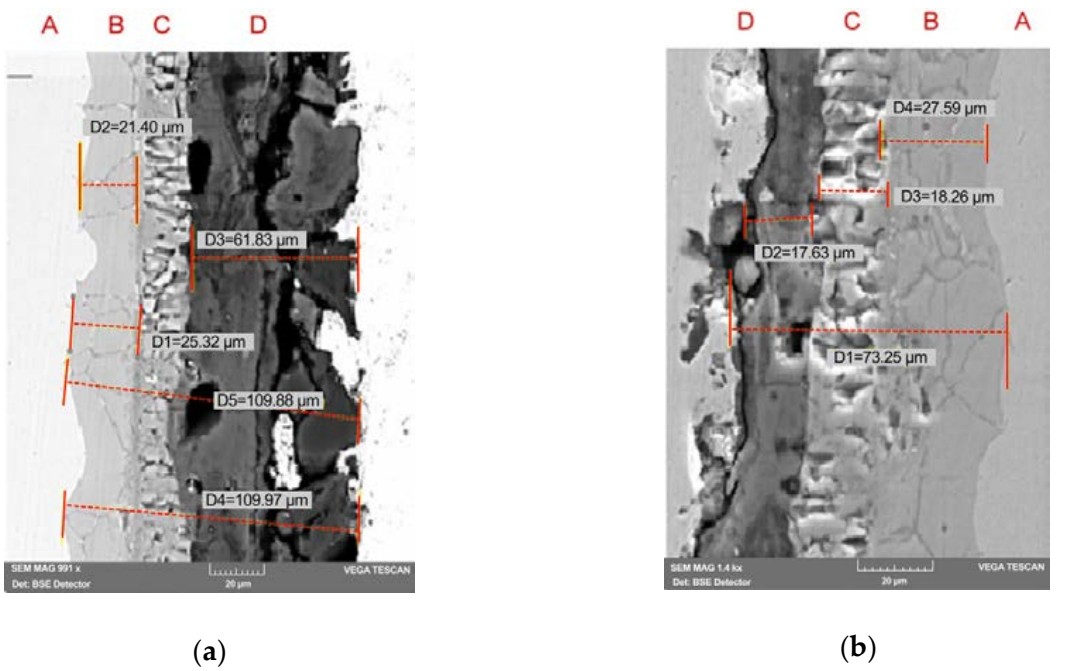

(**a**)                                              (**b**)

**Figure 6.** Cross-sectional SEM images for: (**a**) S1 sample; (**b**) S2 sample.

Figure 6 highlights the layered structure obtained on the 316L austenitic steel substrate (A) by the applied treatment (autoclaving followed by PEO); on the substrate (A), a first layer of oxide ($\approx$ 20–30 μm) (B), followed by a second layer (C) of iron oxide ($\approx$15–20 μm) that develops after autoclaving; After this, following the PEO treatment, a ceramic structure ($\approx$ 20–50 μm) (D) develops, which has an internal area that has a compact appearance and an unregulated outer area with open pores.

According to the literature [18,39], it was generally accepted that the oxide film formed on stainless steels in high temperature water is mainly composed of outer irregular shaped layer made of crystallites rich in iron, and the inner layer with fine-grained oxide, compact and very adherent to base metal, non-porous, very protective, and chromium rich. Numerous studies [40–42] have reported the duplex

oxide layer and describe these crystallites as spinels, which are of the $AB_2O_4$ crystallite type (A = Fe(II), B = Fe(III) or Cr(III)).

Figure 6 clearly shows the duplex structure of the oxide film developed by autoclaving: the inner layer B and the outer layer C.

Figure 7 and Table 4 show the result of EDS analysis at selected points in the sample sections S1 and S2 (in the sample S1, P1 is in the substrate, P2 in the adjacent oxide layer, P3 in the layer of iron oxide, P4 and P5 in the ceramic layer; in the sample S2, P1 is in the substrate, P2, P3 are in the adjacent oxide layer, P4 in the layer of iron oxide, P5 in the ceramic layer).

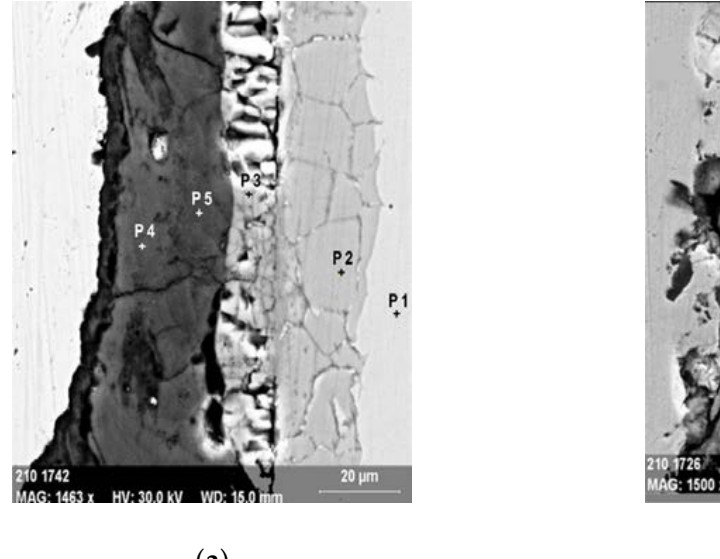

| (a) | (b) |

**Figure 7.** EDS analysis at selected points in the sample sections: (**a**) S1 sample; (**b**) S2 sample.

**Table 4.** EDS results on selected points in the sample sections.

| Sample | Point | Elements [atom.%] | | | | | | |
|---|---|---|---|---|---|---|---|---|
| | | O | Al | Cr | Mn | Fe | Ni | Mo |
| | P1 | - | - | 18.2 | 1.6 | 68.1 | 8.6 | 3.5 |
| | P2 | 9.4 | - | 26.4 | - | 47.0 | 13.8 | 3.4 |
| S1 | P3 | 19.3 | - | 0.7 | 0.4 | 79.6 | - | - |
| | P4 | 30.5 | 36.9 | - | - | 32.6 | - | - |
| | P5 | 31.7 | 45.9 | - | - | 22.4 | - | - |
| | P1 | - | - | 18.0 | 1.3 | 70.1 | 8.2 | 2.4 |
| | P2 | 6.5 | - | 21.2 | - | 57.7 | 14.6 | - |
| S2 | P3 | 4.9 | - | 25.4 | - | 52.7 | 17.0 | - |
| | P4 | 17.5 | - | - | 1.3 | 81.2- | - | - |
| | P5 | 34.4 | 36.9 | - | - | 28.7 | | - |

*3.4. Corrosion Behavior*

Figure 8 shows the polarization curves for 316L samples (S1, S2, and untreated 316L autoclaved) in 0.5M NaCl aqueous electrolyte. The results obtained by the Tafel slope method [43] are shown in Table 5.

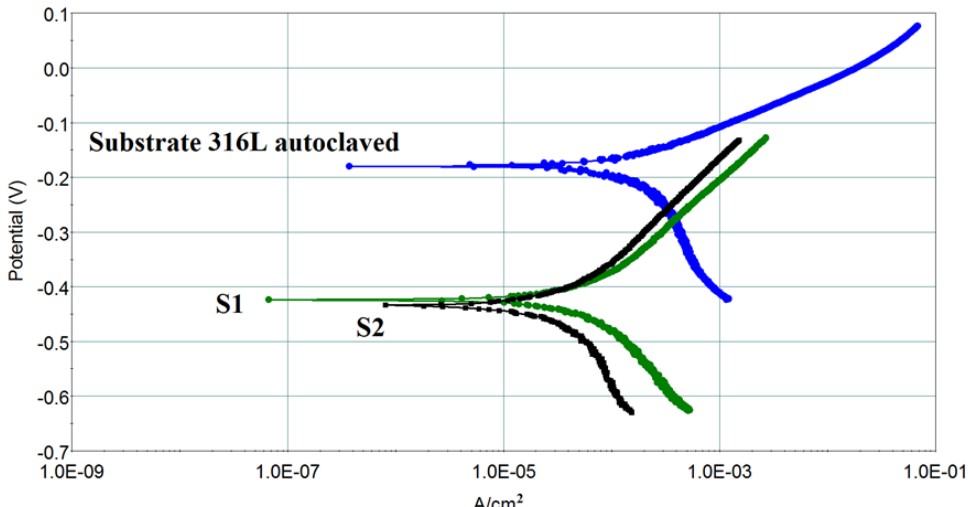

**Figure 8.** Polarization curves for 316L samples (S1, S2, and untreated 316L autoclaved) in 0.5M NaCl aqueous electrolyte.

**Table 5.** Corrosion potential and corrosion rate as resulted from the Tafel slopes method.

| Parameter [Measurement Unit] | 316L | S1 | S2 |
|---|---|---|---|
| $E_{corr}$[1] [mV] | −180 | −424 | −434 |
| $i_{corr}$[2] [μA/cm$^2$] | 129.6 | 50.06 | 112.6 |
| $V_{corr}$[3] [mmpy] | 0.38 | 0.147 | 0.331 |

[1] $E_{corr}$ – corrosion potential; [2] $i_{corr}$ – corrosion current density; [3] $V_{corr}$ – corrosion rate.

The potentiodynamic polarization measurements in 0.5M NaCl aqueous electrolyte (Figure 8), show that the corrosion potential, in the case of samples S1, S2, occurs to more negative values after treatment, but the values of corrosion rate show that the aluminum-based thin films obtained some corrosion protection; a decrease in corrosion currents caused by deposition of aluminum oxide films at micro-arc conditions is limited by the high porosity of film.

The difference in corrosion behavior between samples S1 and S2 reflects a difference between the composition of coating S1 and coating S2: the surface of sample S1 is covered with aluminum oxide, while on the surface of sample S2 there is aluminum oxide, but also a small amount iron oxide. The treatment conditions for the two samples differ, applied effective voltage U = 260V for S1 and U = 220V for S2 (Table 1).

## 4. Discussion

The oxide layer developed on the 316L substrate (zone A, Figure 6) has a duplex structure consisting of:

- an inner layer of oxide adjacent to the substrate of a layer, with fine granulation, non-porous, adherent (zone B, Figure 6); EDS analysis at points in the film section show that this oxide layer is rich in Cr (the analysis in point P2, Figure 7a, and points P2, P3, Figure 7b) show atomic concentrations of Cr in the 21- domain. 27% atom).
- an external layer of Fe oxide, porous (zone C, Figure 6), very rich in Fe (the analysis in point P3, Figure 7a, and point P4, Figure 7b) show atomic concentrations of Fe in range 79–82 atom%).

These results are in agreement with the literature [18,39]. In the literature [40–42], the external oxide layer is described as consisting of spins of type $AB_2O_4$ (A = Fe (II), B = Fe (III), or Cr (III)).

The outer layer of the structure developed on the 316L substrate by autoclaving, fulfills the role of the barrier layer required for initiation of PEO process. During the PEO process oxygen consumption

takes place (through the release of O2 in the stage prior to entering the stable discharge regime and through the oxidation process); this explains the oxygen depletion of the outer layer of porous Fe oxide, which does not correspond to a formula of type AB2O4 (A = Fe (II), B = Fe (III), or Cr (III)), the content of Fe being much more high.

Our results obtained by XPS, XRD, SEM, EDS, show that the PEO treatment applied to the autoclaved 316L samples under the conditions described leads to the formation of aluminum oxide ceramic coatings with thicknesses in the range of 20–50 μm, with a porous appearance, containing $Al_2O_3$, $FeAl_2O_4$, $Fe_2O_3$, and Fe. The superficial layer (50 Å depths) is composed of Al oxide.

It is necessary to compare our results with those of the literature concerning similar treatment applied to austenitic stainless steel 316L [29]. According to Wu et al. [29], a DC power supply was used, constant voltage conditions, 320 V, 3 min, $T_{electrolyte}$ < 45 °C.

Following this treatment, a ceramic coating was obtained composed of Al oxides (prevailing), Fe oxides, and Al hydroxide (thickness ≈ 30 μm), the percentage of $Al_2O_3$ on the surface being twice as low as the treatment described in this paper.

Much attention is paid to the corrosion resistance and abrasion resistance of alumina coatings. However, these performances are closely related to the content of $\alpha$-$Al_2O_3$ in coatings [27].

If the oxidation process in electrolytic plasma occurs by applying a continuous voltage, then it is very sensitive to the applied voltage, and the range for which the spark and micro arc discharge process is very narrow. If the applied voltage deviates slightly from the values in this range, then the process ceases or goes from electric discharge into the arc and the deposited layer is destroyed.

The process is more stable if unipolar voltage pulses are applied to the sample. In this case, the spark and micro arc discharge process is stable over a much higher voltage range, reaching values that allow for thick layers. The realization of short-wave arcs in the short arc due to voltage impulses, prevents them from passing into electric arc discharge, does not allow the thermal destruction of deposited layers and the formation of defects in them.

There are two significant differences between continuous and pulse modes PEO:

- the voltage in the direct current PEO mode (320 V in [30,32]) is applied permanently on the sample, while in PEO pulse mode, a time interval of one period is applied, after which it is zero. In our experiment, the pulse voltage applied is over 500 V, much higher than in DC, which allows for higher temperatures in electrical discharge in the microarc and the synthesis of a larger amount of $Al_2O_3$;
- between the two impulses, the oxidation layer material cools and crystallizes the molten portions of the layer, which makes it possible to obtain layers of oxide with much higher thickness than in the continuous PEO mode.

Polarization curve shows that a decrease in corrosion currents caused by deposition of aluminum oxide films at micro-arc conditions, in 0.5 M NaCl aqueous electrolyte, at room temperature, is limited by the high porosity of film.

Austenitic steels suffer from severe corrosion attack in lead or LBE melt at temperatures above 500 °C. Obtaining some $Al_2O_3$ coatings on 316L steel takes into account the improvement of the corrosion resistance of stainless steels exposed to oxygen—containing HLM at temperatures above 500 °C. The use of austenitic 316L as a nuclear material for nuclear systems with Heavy Liquid Metals (HLM) as lead or Lead Bismuth Eutectic (LBE), requires Al-based ceramic-based coatings with controlled properties (high $Al_2O_3$ content, thickness greater than 30 microns, high adhesion) [4,12]. The obtained results show that it is feasible to obtain by PEO ceramic coatings of Al oxide on 316L steel suitable for use as structural material in the nuclear field.

316L steel generally exhibits good corrosion behavior. Coatings with $Al_2O_3$ are of interest for improving the corrosion behavior in the case of prolonged exposure at temperatures close to 700 °C in aggressive environments, for example, in the use in the metallurgical industry, or in the nuclear industry.

Measurements by the Tafel slope method, performed at room temperature, show an improvement of the corrosion behavior (lower values of corrosion current on the treated samples, compared to the

untreated sample) but a decrease of the corrosion currents caused by the deposition of aluminum oxide films. Under micro-arc conditions it is limited by the high porosity of the film.

For the use of 316L steel in conditions of long exposure at temperatures close to 700 °C in aggressive environments, starting from the preliminary results presented in this paper, we propose to approach the following steps:

- development of easier pre-treatments for achieving the barrier layer necessary to start PEO processes, which will replace the autoclaving stage; we consider cathodic oxidation treatments in electrolysis plasma [44];
- the realization of the PEO treatments on the samples submitted prior to the pretreatment for the achievement of the barrier layer, following: the fulfillment of the conditions for the treatment of some large samples; clogging of the pores of the surface film developed by PEO by electroplating with different nanoparticles, as recommended in [45];
- testing the electrochemical behavior of the obtained structures, under test conditions that simulate the operating conditions (electrochemical autoclaves, circulation loops).

## 5. Conclusions

We proved the feasibility of using Plasma Electrolytic Oxidation in aqueous solution of $NaAlO_2$ using a pulsed unipolar power supply for preparation of ceramic–like aluminum-oxide films, with thicknesses in the range 20–50 μm, with high content of $Al_2O_3$ on the surface of austenitic stainless steels.

Electrochemical tests show that although the ceramic coating obtained is porous, it does not worsen the corrosion behavior, in 0.5 M NaCl aqueous electrolyte, at room temperature; the values of corrosion rate show that the aluminum-based thin films obtained ensure some corrosion protection.

The obtained results show that it is feasible to obtain by PEO ceramic coatings of Al oxide on 316L steel, suitable for use in aggressive environments at high temperatures (e.g., in boilers and furnaces, in nuclear installations as structural material).

**Author Contributions:** Conceptualization, V.A.A., C.R., and V.M.; methodology, A.M., E.C., and M.M.; software, I.A.B.; validation, V.A.A., C.R., and V.M.; investigation, A.M., E.C., M.M., C.N.M., I.D.D., S.T., and I.A.B.; writing—original draft preparation, V.A.A., C.R., and V.M.; writing—review and editing, V.A.A., C.R., V.M., I.D.D., and I.A.B.; visualization, I.A.B. and I.D.D; supervision, C.R.; project administration, V.A.A. and C.R.; funding acquisition, V.A.A. and C.R. All authors have read and agreed to the published version of the manuscript.

**Funding:** This research was funded by Bilateral Project 04-4-1121-2015/2020, between Valahia University of Targoviste and Joint Institute for Nuclear Research, Dubna, Moscow Region; Protocol 4755-4-2018/2020, Micro-structural and compositional characterization of supports and coating layers on different substrates applied in biomaterials, photoelectrochemicals catalysis and cultural heritage.

**Conflicts of Interest:** The authors declare no conflicts of interest.

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
