# Peer review of "Aluminum Oxide Ceramic Coatings on 316l Austenitic Steel Obtained by Plasma Electrolysis Oxidation Using a Pulsed Unipolar Power Supply"

_coatings, doi:10.3390/coatings10040318_

Round 1
Reviewer 1 Report
Authors did good work, this work is very useful for immediate engineering applications. this kind of work should be encouraged to publish for the benefit of many engineers to use this process directly for their applications to enhance the efficiency, safety, life of the component, etc. However, following a few comments or suggestions may be observed to improve the quality of the manuscript as well to make it more reader-friendly.
1) Keywords - no abbreviations, write in full form
2) The test sample is so small 20x15x1 mm3 - is it possible to demonstrate in big size? what is the feasibility for making in big size complex shapes, to be addressed?
3) Experimental section - for all instruments or equipment used to be quoted in the following standard order "(model, make, city, country)".
4) Fig. 1 - some unidentified peaks were seen in 1b. Fig. 1a and 1b keep in a single plot similar to Fig. 4 with more space between two plots.
5) It is good to add XPS data for the SS sample before PEO treated - for comparison sake.
6) Fig. 2 can be combined with Fig. 1 as an inset of Al2O3 peaks. For other spectrums of Cr, Ni, Fe need not present, as anything to see in those images.
7) all quantification data from all characterisation techniques can be compiled in a single table, it looks good and eases for comparison.
8) Fig. 4 - mention JCPDS number in the figure itself along with labels, instead of in the text. Fig. 4 - please keep the gap between two plots (along the y-axis) for better visibility.
9) Fig. 5 - only high-resolution images of S1 and S2 are sufficient, remaining images present as nothing to explain or visible any features. It is good to add one SS sample before PEO treatment.
10) Fig. 6 - Actual SEM image missed - that also to be presented. Add for pre-PEO sample also. b) and d) data can be shifted to the Table as suggested above.
11) All results are presented along with figures, but nothing much explained. it is good, if elaborate in detail.
12) Fig. 7 observations as indicated in the text - no evidence for formed iron oxide or different features. explain with evidence or some references.
13) Fig. 7 and 8 keep all figures in the same size. 8a & 8 b data can be compiled with the consolidation table as suggested above. Label all figures with important features appropriately. Indicate the cross-section of the sample, where are the base and top surface - indicate those features in the figure itself to make it more reader-friendly.
14) Fig. 9 - compared the data with the pre-PEO sample, so explain the corrosion resistance behaviour more effectively, instead of in Table 2.
15) Why wide variation of values for S1 and S2 (Table 2)? although both samples are coated with alumina.
Author Response
Reviewer 1
1) Keywords - no abbreviations, write in full form
The keywords were written without abbreviations. (page 1, lines 32-34 )
2) The test sample is so small 20x15x1 mm3 - is it possible to demonstrate in big size? what is the feasibility for making in big size complex shapes, to be addressed?
In the revised form of the article was inserted:
“Compared with other techniques that can be used to make protective ceramic coatings on austenitic steels (the thermal spraying technique and the MIEB-Al procedure), PEO is cheaper and more suitable for treating complex geometry samples.”(page 3, lines 100-102)
“An advantage of the PEO technique is that it allows the treatment of samples with different shapes and sizes; the important parameter is the current density, the treatment of large samples requiring the use of high power sources. The dimensions of the samples used in this study are suitable for the development of the treatment method, being suitable for performing different analyzes to characterize the coatings made.” (page 3, lines 142-143; page 4, lines 144-146).
3) Experimental section - for all instruments or equipment used to be quoted in the following standard order "(model, make, city, country)".
In the Experimental Section, for all instruments or equipment were inserted missing data (page 4, lines 170-171; page 4 lines 184-185; page 5 line 190; page 5 lines 194-195; page 5 lines 198-200)
4) Fig. 1 - some unidentified peaks were seen in 1b. Fig. 1a and 1b keep in a single plot similar to Fig. 4 with more space between two plots.
Fig. 1a and 1b were combined in a single plot – Fig. 1(b)(page 6, lines 213-217)
5) It is good to add XPS data for the SS sample before PEO treated - for comparison sake.
The XPS survey spectra of 316L sample autoclaved before PEO was inserted in revised article – Fig. 1(a) (page 5, lines 211-212)
6) Fig. 2 can be combined with Fig. 1 as an inset of Al2O3 peaks. For other spectrums of Cr, Ni, Fe need not present, as anything to see in those images.
In the revised article was inserted:
“Fig. 1(a-e) show clearly that on the surface of the sample S1 are not present the elements Fe, Cr, Ni but only aluminum oxide, and Fig. 2(a-e), clearly show that on the surface of the sample S2 are not present the elements Cr, Ni but only aluminum oxide - majority and Fe oxide.” (page 7, lines 226-229)
7) All quantification data from all characterisation techniques can be compiled in a single table, it looks good and eases for comparison.
The quantification data presented in Figure 6 and Fig. 8 were compiled in Table 2 (EDS results on S1 and S2 samples) (page 10, line 262) and Table 3 (EDS results on selected points in the sample sections) (page 11, line 284)
8) Fig. 4 - mention JCPDS number in the figure itself along with labels, instead of in the text. Fig. 4 - please keep the gap between two plots (along the y-axis) for better visibility.
In Fig. 4 were inserted the JCPDS numbers (page 8, line 240). The software does not allow keeping the gap between two plots.
9) Fig. 5 - only high-resolution images of S1 and S2 are sufficient, remaining images present as nothing to explain or visible any features. It is good to add one SS sample before PEO treatment.
In Fig. 5 was inserted the SEM image obtained on 316L sample autoclaved before PEO – Fig. 5(a) (page 8, line 246).
For S1 and S2 samples were kept only the SEM images with high resolution – Fig. 5 (b) and Fig. 5 (c). (page 9, lines 247-248)
10) Fig. 6 - Actual SEM image missed - that also to be presented. Add for pre-PEO sample also. b) and d) data can be shifted to the Table as suggested above.
In Fig. 6 (a) & (b) were inserted the SEM image, but for 316L sample autoclaved before PEO unfortunately the authors missed the initial data. The quantification data presented in Figure 6 were compiled in Table 2. (page 10, line 262)
11) All results are presented along with figures, but nothing much explained. it is good, if elaborate in detail.
In the revised article was inserted:
“The oxide layer developed on the 316L substrate (zone A, Fig. 7) has a duplex structure consisting of:
- an inner layer of oxide adjacent to the substrate of a layer, with fine granulation, non-porous, adherent (zone B, Fig.7); EDS analyzes at points in the film section show that this oxide layer is rich in Cr (the analyzes in point P2, Fig.8a, and points P2, P3, Fig. 8b) show atomic concentrations of Cr in the 21- domain. 27% atom).
- an external layer of Fe oxide, porous (zone C, Fig. 7), very rich in Fe (the analyzes in point P3, Fig. 8a, and point P4, Fig. 8b) show atomic concentrations of Fe in range 79-82 atom%).
These results are in agreement with the literature [18,39]. In literature [40-42] the external oxide layer is described as consisting of spins of type AB2O4 (A = Fe (II), B = Fe (III) or Cr (III)).” (page 12, lines 312-321)
12) Fig. 7 observations as indicated in the text - no evidence for formed iron oxide or different features. explain with evidence or some references.
In the revised article was inserted:
“During the PEO process oxygen consumption takes place (through the release of O2 in the stage prior to entering the stable discharge regime and through the oxidation process); this explains the oxygen depletion of the outer layer of porous Fe oxide, which does not correspond to a formula of type AB2O4 (A = Fe (II), B = Fe (III) or Cr (III)), the content of Fe being much more picked up.” (page 13, lines 323-327)
13) Fig. 7 and 8 keep all figures in the same size. 8a & 8 b data can be compiled with the consolidation table as suggested above. Label all figures with important features appropriately. Indicate the cross-section of the sample, where are the base and top surface - indicate those features in the figure itself to make it more reader-friendly.
The quantification data presented in Fig. 8 were compiled in Table 3 (page 11, lines 284)
In the revised article is mention: “Fig. 7 highlights the layered structure obtained on the 316L austenitic steel substrate (A) by the applied treatment (autoclaving followed by PEO); on the substrate (A), a first layer of oxide (≈ 20-30 μm) (B), followed by a second layer (C) of iron oxide (≈15-20 μm) develops after autoclaving; Over this, following the PEO treatment, a ceramic structure (≈ 20-50 μm) (D) develops, which has an internal area that has a compact appearance and an unregulated outer area with open pores.” (page 10, lines 268-272)
14) Fig. 9 - compared the data with the pre-PEO sample, so explain the corrosion resistance behaviour more effectively, instead of in Table 2.
Due to the new 2 Tables inserted in the revised article, Table 2 is Table 4.
In the revised article were inserted several explanations:
“316L steel generally exhibits good corrosion behavior. Coatings with Al2O3 are of interest for improving the corrosion behavior in the case of prolonged exposure at temperatures close to 700 °C in aggressive environments, for example, in the use in the metallurgical industry, or in the nuclear industry.
Measurements by the Tafel slope method, performed at room temperature, show an improvement of the corrosion behavior (lower values of corrosion current on the treated samples, compared to the untreated sample) but a decrease of the corrosion currents caused by the deposition of aluminum oxide films. under micro-arc conditions it is limited by the high porosity of the film.
For the use of 316L steel in conditions of long exposure at temperatures close to 700 °C in aggressive environments, starting from the preliminary results presented in this paper, we propose to approach the following steps:
-development of easier pre-treatments for achieving the barrier layer necessary to start PEO processes, which will replace the autoclaving stage; we consider cathodic oxidation treatments in electrolysis plasma [44];
- the realization of the PEO treatments on the samples submitted prior to the pretreatment for the achievement of the barrier layer, following: the fulfillment of the conditions for the treatment of some large samples; clogging of the pores of the surface film developed by PEO by electroplating with different nanoparticles, as recommended in [15];
- testing the electrochemical behavior of the obtained structures, under test conditions that simulate the operating conditions (electrochemical autoclaves, circulation loops).” (pages 13-14, lines 369-388)
15) Why wide variation of values for S1 and S2 (Table 2)? although both samples are coated with alumina.
In the revised article were inserted several explanations:
“The difference in corrosion behavior between samples S1 and S2 reflects a difference between the composition of coating S1 and coating S2: the surface of sample S1 is covered with aluminum oxide, while on the surface of sample S2 there is aluminum oxide, but also a small amount of oxide of Faith. The treatment conditions for the two samples differ, applied effective voltage U = 260V for S1 and U = 220V for S2 (Table 1).” (page 12, lines 305-309)
“The oxide layer developed on the 316L substrate (zone A, Fig. 7) has a duplex structure consisting of:
- an inner layer of oxide adjacent to the substrate of a layer, with fine granulation, non-porous, adherent (zone B, Fig.7); EDS analyzes at points in the film section show that this oxide layer is rich in Cr (the analyzes in point P2, Fig.8a, and points P2, P3, Fig. 8b) show atomic concentrations of Cr in the 21- domain. 27% atom).
- an external layer of Fe oxide, porous (zone C, Fig. 7), very rich in Fe (the analyzes in point P3, Fig. 8a, and point P4, Fig. 8b) show atomic concentrations of Fe in range 79-82 atom%).
These results are in agreement with the literature [18,39]. In literature [40-42] the external oxide layer is described as consisting of spins of type AB2O4 (A = Fe (II), B = Fe (III) or Cr (III)).” (page 12, lines 312-321)
Reviewer 2 Report
The authors have presented a simple but useful study into the fabrication of alumina coatings on steel substrates.
This work may be of interest to those working in the field of such coatings and therefore merits publication.
Before the work is published the authors should make some efforts to:
- Raise the quality of English in the manuscript
- Clarify the rationale and reasons for the methods used
- Compare this work with alternative methods for Alumina coatings on metal
- Explain how others can use this work to design materials in the future
Author Response
Reviewer 2
- Raise the quality of English in the manuscript
The entire manuscript was revised from English language and context point of view by a native professor (with PhD in particular field). The Certificate of Translation was uploaded to the Editor.
- Clarify the rationale and reasons for the methods used
In the revised article was inserted: “In order to understand the processes of electrolytic oxidation and to develop the experimental method for making ceramic coatings based on aluminum oxide on a stainless steel substrate, it is necessary to characterize the deposits in terms of morphology, structure and composition both in the surface area and in the bulk deposition; the properties of the surface area determine the corrosion behavior of the material.” (page 4 lines 164-168)
- Compare this work with alternative methods for Alumina coatings on metal
In the revised article was inserted: “For the use of 316L steel in conditions of long exposure at temperatures close to 700 ° C in aggressive environments, starting from the preliminary results presented in this paper, we propose to approach the following steps:
-development of easier pre-treatments for achieving the barrier layer necessary to start PEO processes, which will replace the autoclaving stage; we consider cathodic oxidation treatments in electrolysis plasma [44];
- the realization of the PEO treatments on the samples submitted prior to the pretreatment for the achievement of the barrier layer, following: the fulfillment of the conditions for the treatment of some large samples; clogging of the pores of the surface film developed by PEO by electroplating with different nanoparticles, as recommended in [45]” (page 14 lines 376-385)
- Explain how others can use this work to design materials in the future
In the revised article were inserted several explanations:
“316L steel generally exhibits good corrosion behavior. Coatings with Al2O3 are of interest for improving the corrosion behavior in the case of prolonged exposure at temperatures close to 700 ° C in aggressive environments, for example, in the use in the metallurgical industry, or in the nuclear industry.
Measurements by the Tafel slope method, performed at room temperature, show an improvement of the corrosion behavior (lower values of corrosion current on the treated samples, compared to the untreated sample) but a decrease of the corrosion currents caused by the deposition of aluminum oxide films. under micro-arc conditions it is limited by the high porosity of the film.
For the use of 316L steel in conditions of long exposure at temperatures close to 700 ° C in aggressive environments, starting from the preliminary results presented in this paper, we propose to approach the following steps:
-development of easier pre-treatments for achieving the barrier layer necessary to start PEO processes, which will replace the autoclaving stage; we consider cathodic oxidation treatments in electrolysis plasma [44];
- the realization of the PEO treatments on the samples submitted prior to the pretreatment for the achievement of the barrier layer, following: the fulfillment of the conditions for the treatment of some large samples; clogging of the pores of the surface film developed by PEO by electroplating with different nanoparticles, as recommended in [45];
- testing the electrochemical behavior of the obtained structures, under test conditions that simulate the operating conditions (electrochemical autoclaves, circulation loops).” (pages 13-14, lines 368-387)